# *Trichocladium solani* sp. nov.—A New Pathogen on Potato Tubers Causing Yellow Rot

**DOI:** 10.3390/jof8111160

**Published:** 2022-11-02

**Authors:** Arseniy Belosokhov, Maria Yarmeeva, Lyudmila Kokaeva, Elena Chudinova, Svyatoslav Mislavskiy, Sergey Elansky

**Affiliations:** 1Faculty of Biology, Department of Mycology and algology, Lomonosov Moscow State University, Leninskiye Gory 1, Building 12, 119991 Moscow, Russia; 2Agricultural Technology Institute, Peoples Friendship University of Russia (RUDN University), Miklukho-Maklaya st. 8, Building 2, 117198 Moscow, Russia

**Keywords:** Chaetomiaceae, dry rot, *Fusarium*, pathogen, potato, *Trichocladium*, tubers, yellow rot, one new species

## Abstract

A new species, *Trichocladium solani*, was isolated from potato (*Solanum tuberosum* L.) tubers from Russia. The species has no observed teleomorph and is characterized morphologically by non-specific *Acremonium*-like conidia on single phialides and chains of swollen chlamydospores. Phylogenetic analysis placed the new species in a monophyletic clade inside the *Trichocladium* lineage with a high level of support from a multi-locus analysis of three gene regions: ITS, *tub2*, and *rpb2*. ITS is found to be insufficient for species delimitation and is not recommended for identification purposes in screening studies. *T. solani* is pathogenic to potato tubers and causes lesions that look similar to symptoms of *Fusarium* dry rot infection but with yellowish or greenish tint in the necrotized area. The disease has been named “yellow rot of potato tubers”.

## 1. Introduction

Pathogenic fungi are prominent agents in ecosystems, involved in the control of populations of plants in natural ecosystems [1] or causing potentially significant damage to agricultural ecosystems (www.fao.org (accessed on 20 August 2022)), and they require control measures for crops’ protection. For effective control, it is widely accepted that the identification of a causative agent is one of the first and most important steps to ensure appropriate crop protection actions are implemented [2,3]. Fungal identification is known to be a demanding task. Hyphomycetes often show little difference in their morphology, producing similar propagules or chlamydospores, and sometimes do not produce teleomorphs or anamorphs, or neither, appearing sterile. Phylogenetic studies using DNA sequencing provide vital clues to what phenotypic features might be informative and serve as barcodes for identification purposes. Most identification is performed using ITS sequence data which is a well-established barcode for the identification of most fungi [4,5,6]. However, when observing ambiguous, poorly featured isolates from crop-related material, it is crucial to carry out a careful and thorough investigation as novel pathogens emerge continuously [7,8] and their timely identification and monitoring are at the base of any prevention measures.

Chaetomiaceae is a diverse family of fungi with more or less well-developed distinctive perithecial ascomata with a hairy surface developing terminal and/or lateral hairs of various forms. Their anamorphs vary significantly but are sometimes supplemented with nonspecific *Acremonium*-like conidiation with single phialides producing chained conidia [9,10,11]. Some species are known to be thermophilic [11,12,13], some are able to decompose cellulose [14], and some are promising agents of biocontrol of other pathogenic fungi [15,16]. The identification of species within the group is mostly based on the morphology of sexual structures that show robust delimitation of species that is predominately in line with phylogenetic predictions [10,11,13]. However, when the teleomorph is not observed, additional steps are required to identify an isolate. While ITS and LSU regions are widely used for identification, the phylogenetic signal in the regions is low within Chaetomiaceae, and these genes show little sufficiency for delimiting genera [13], let alone species. Hambleton et al. [5] noted that the ITS sequences of *Humicola grisea* and *Trichocladium asperum*, type species of the genus, were identical, which makes the delimitation impossible using internal transcribed spacers. As an example, *Humicola koreana* was described using ITS and LSU sequencing and compared only with *Humicola* species. Later, Wang at al. [13] redefined the species, placing it within the *Staphylotrichum* using *tub2* and *rpb2* in addition to ITS.

The genus *Trichocladium* is a diverse group within the Chaetomiaceae that is well-defined phylogenetically; however, it is not straightforward to outline morphologically. Morphological features of species within the genus are not specific to *Trichocladium*. The genus in the current sense [13] includes species that were previously placed, due to their morphology, within *Chaetomidium*, *Chaetomium*, *Thielavia*, *Beniowskia*, *Gilmaniella*, *Humicola*, and *Monodictys*, asexual members encompassing a significant heterogeneity in conidial shape, pigmentation, and development. Wang et al. [13] described the genus as having four subclades: Subclade 1 included the lectotype species *T. asperum*, which produced pigmented, solitary, monoblastic conidia on somewhat differentiated conidiophores. Subclades 2 and 3 contained only sexual species that produced ostiolate or non-ostiolate ascomata. Species in subclade 4 produced hyaline asexual structures (conidia), or only poorly differentiated structures, with sexual stages present or absent. There are not a lot of specific data for the ecology of *Trichocladium* species. The most featured mention in the literature reports that the species are frequently associated with submerged decomposing wood in tropical areas [17] or isolated from dead plant material and associated substrata [13], and a few members are known to be thermophilic [17].

Potato (*Solanum tuberosum* L.) is one of the key agricultural crops with the volume of a total global production of 359 million tons in 2020 [18]. Being the third most consumed crop in the world after rice and wheat, potato crop is susceptible to many biotic and abiotic diseases [3,19]. Currently, a number of potato tuber infections are considered economically important and include galls (powdery scab, wart, common scab, etc.), blemishes (black dot, black scurf, skin spot silver scurf, etc.) and rots (charcoal rot, bacterial soft rot, black leg, pink rot, ring rot, brown rot, *Fusarium* dry rot, etc.) [3]. Because of simultaneous environmental challenges including viral, bacterial, and fungal diseases and abiotic factors affecting potato, it is tough to estimate the exact loss of yield due to fungal infections [20]. However, it is a common understanding that unless appropriate measures are undertaken beforehand, fungal infections of potato can cause significant loss of yield [21,22] resulting in potentially devastating consequences [23]. With new fungal pathogens being reported regularly [24,25,26,27,28,29,30], it is essential to monitor and control diseases causing yield losses and decreasing the quality of affected tubers.

In this study, we aimed to analyze several isolates of *Trichocladium* obtained from infected tubers of *Solanum tuberosum* collected in 2017–2019 from potato storages. The main goal was to describe morphological and phylogenetic features as well as testing for pathogenicity towards potato tubers.

## 2. Materials and Methods

***Sampling and isolation of cultures***—Samples of infected potato (*Solanum tuberosum* L.) tubers or various potato cultivars were collected from commercial potato fields or storages in different regions of the Moscow and Kaluga regions in Russia (Table 1), with each sample consisting of 8–10 tubers with 10–30% of visible surface damage. External growing conditions for collected tubers, such as irrigation scheduling, fungicide treatment, crop rotation patterns, and concomitant infections at each collection site varied. The storing conditions of sampled tubers from different storages were 6–10 °C and 85–90% humidity. Infected potato tubers were thoroughly washed and then submerged into a sanitizing solution of 2% sodium hypochlorite for the removal of surface contamination before being air-dried. The tubers were then sliced across the damaged areas with a sterile blade to remove the necrotized tissue. A slice of living infected potato tissue under the necrosis was removed and placed on plates with potato dextrose agar (PDA) [31] in sterile conditions. In order to collect the air sample, opened PDA plates were exposed for one hour inside the sorting compartment of a potato storage facility, closed, and then cultured and purified from contamination using standard culture methods. All the isolates used in this study are listed in Table 1, including the obtained sequence data generated in [13].

*DNA isolation, sequencing, and phylogeny*—Genomic DNA was extracted from fungal mycelium grown on PDA after 6 days of growth. The mycelium was placed into 2.0 mL sterile reinforced microtubes with caps (SSI, USA) along with 700 μL of CTAB buffer (1.4 M NaCl, 0.1 M Tris-HCl, 20 mM EDTA, and 2% hexadecyltrimethylammonium bromide) and ground with zirconium oxide beads (Bertin Instuments, Montigny-le-Bretonneux, France) in the following composition: 1 × 5 mm; 2 × 2 mm; 3 × 0.1 mm (per tube) using the Precellys^®^ Evolution homogenizer (Bertin Instruments, Montigny-le-Bretonneux, France) in two cycles of 8000 rpm for 10 s with 5 s resting time in between. Tubes then were incubated at 65 °C for 1 h with intensive shaking every 20 min. After incubation, 500 μL of cold chloroform was added to the tubes that were centrifuged at 16,249× *g* for 10 min, and the supernatants were transferred to clean 1.5 mL microcentrifuge tubes. Isopropanol (400 μL) and 70 μL of 5 M potassium acetate (pH 4.6) were added to each supernatant; then, the tubes were shaken carefully and centrifuged at 16,249× *g* for 10 min at room temperature. The supernatants were then discarded, 150 μL of 70% ethanol (*v/v*) was added, and the tubes were centrifuged at the same gravity force for 5 min twice. Pellets were air-dried at room temperature and resuspended in 50 μL of deionized water. The resulting DNA solutions were diluted to a concentration of ~50 ng/μL. DNA concentration was determined using a spectrophotometer NanoDrop 2000 (Thermo Scientific, Waltham, MA, USA) by measuring absorbance at 260 nm.

DNA amplifications were performed in a 25 μL total volume reaction containing 1 μL of a DNA template (50 ng/μL), 2.5 μL of 10x PCR buffer (Applied Biosystems, Waltham, MA, USA), 1.5 mM MgCl_2_, 120 µM (each) deoxyribonucleotide triphosphates (dNTP), 0.2 µM of each primer (Evrogen Co, Moscow, Russia), 1.5 U of Taq polymerase (Promega, Madison, WI, USA), and 20 μL of deionized water (MQ) for ITS and *tub2* amplifications. For the amplification of *rpb*2, the following mix per tube was used: 10x PCR buffer—2.5 μL; MgCl2—2.1 mM; Taq-polymerase –1.5 U; dNTP (each)—180 µM; 0.34 µM of each primer; MQ—21.76 μL; and Template DNA—37.5 ng.

The following primers were used for PCR amplification: **ITS1** 5′-TCCGTAGGTGAACCTGCGG-’3 and **ITS4** 5′-TCCTCCGCTTATTGATATGC-3′ [4] for the internal transcribed spacer regions (ITS), including 5.8S nrRNA gene region; **T1** 5′-AACATGCGTGAGATTGTAAGT-3′ [32], and **TUB4Rd** 5′-CCRGAYTGRCCRAARACRAAGTTGTC-3′ [33] for the partial beta-tubulin (tub2) gene region; **rpb2-5F2** 5′-GGGGWGAYCAGAAGAAGGC-3′ [34] and **rpb2AM-7R** 5′-CCCATRGCTTGTYYRCCCAT-3′ [35] for the second largest subunit of DNA-directed RNA polymerase II (rpb2) gene region.

The cycle conditions for the amplification of internal transcribed spacer regions (ITS) included cycles of 95 °C/3 min (initial denaturation) followed by 30x cycles of 94 °C/30 s, 55 °C/30 s, 72 °C/45 s, and ending with 72 °C/8 min (final extension). The cycle conditions of the amplification of partial tub2 gene using T1/TUB4Rd were identical to the ITS protocol, apart from the annealing cycle of 60 °C/30 s. The protocol for the amplification of the partial rpb2 gene using rpb2-5F2/rpb2AM-7R included the initial denaturation 95 °C/3 min followed by 5x cycles of 94 °C/30 s, 58 °C/30 s, and 72 °C/60 s; then, 10x cycles of 94 °C/30 s, 56 °C/45 s, and 72 °C/60 s (10x); and 15x 94 °C/30 s, 56 °CΔ − 0.1 °C (each new cycle, the annealing temperature was lowered by 0.1 °C until it reached 54.5 °C)/30 s; 72 °C/60 sΔ + 2 s (each new cycle, the elongation phase was extended by 2 s); ending with the final extension of 72 °C/5 min. The amplification was performed on a T100 Thermal Cycler (Bio-Rad Laboratories Inc., Hercules, CA, USA).

Target PCR products were run in a 1.5% agarose gel stained with ethidium bromide (0.5 μg/mL) in 0.5× Trisborate EDTA (TBE) buffer at a constant voltage of 85V for about 1 h and visualized under ultraviolet (UV) light. PCR products were extracted from agarose gels and cleaned using Cleanup Mini Kit (Evrogen Co, Moscow, Russia), then sequenced using the BigDye^®^Terminator v.3.1 Cycle Sequencing Kit (Applied Biosystems, San Francisco, CA, USA) and the Applied Biosystems 3730 xl automated sequencer (Applied Biosystems, USA). Each fragment was sequenced in both directions using the same primers described above. Consensus sequences for each locus were assembled using GENEIOUS PRIME 2022.2 software and MEGA X [36]. The obtained sequences were aligned with publicly available sequences from [13] using MEGA X software. Phylogenetic analyses were based on Bayesian inference (BI) and maximum likelihood (ML). ML analysis was performed with MEGA X and IQ-TREE web server [37]. For BI, BEAST the best evolutionary model for each locus was determined using MRMODELTEST 2.0 [38] and IQ-TREE [39]. Obtained trees were viewed in FIGTREE 1.1.2 [40] and subsequently visually prepared and edited in ADOBE^®^ ILLUSTRATOR^®^ 22.3.0. 

*Morphology*—Colony morphology was examined on four different media: PDA, oatmeal agar (OA), cornmeal agar (CMA), malt extract agar (MEA), and potato carrot agar (PCA) prepared as described by Crous et al. [31]. Cultures were inoculated in a one-point fashion and incubated in the dark at 25 °C and 37 °C. Colony diameters were measured after 7 d. To trigger teleomorph formation, several measures were undertaken: colonies growing on PDA at 25 °C were exposed to UV-A light [41,42] for 12 h and then incubated at the same temperature for another 7 d; heat-shock treatment of colonies on OA incubated at 25 °C subjected to 42 °C for 2 h [43]; growth on Hutchinson medium (g/L: KH_2_PO_4_—0.1; NaCl—0.1; CaCl_2_—0.1; FeCl_3_—0.1; MgSO_4_·7H_2_O—0.3; NaNO_3_—2.5; agar—20) [44] with different cellulose substrates as a source of carbon: cellophane, ashless pulp, sulfated (kraft) cellulose, straw, oak and maple shavings, and pine chips for 6.5 months at 25 °C.

Morphological descriptions are based on cultures grown on OA and PDA. Microscopic observations were performed in lactic acid mounts that were gently heated to remove air bubbles or on thin agar blocks with the mycelium that were cut from the Petri plate to the size of the cover glass and rinsed with a detergent solution to remove excessive conidia, if necessary. Alternatively, transparent adhesive tape was used for the preparation of the mount [45]. At least 35 measurements were made for all morphologically informative features. The measurements include the extreme values given in parentheses and, in between, the 95% confidence interval of 30 individual measurements using the methodology described in [10].

Whereas a common practice is to describe culture characteristics with colors using *A mycological colour chart* by R.W. Rayner [46], the color pallets can vary depending on the condition of the paper print or scanned material and the publication itself is not always available for some research groups. Therefore, additional color measurements were developed and implemented to provide more constant data that can be reproduced regardless of the availability of a specific color chart. In order to create the reference color pallet plates with grown culture, they were photographed with SpyderCHECKR 24 charts (Datacolor, USA). The photos were then white-balanced and light-balanced on the color checker using ADOBE^®^ PHOTOSHOP^®^ 2022 23.1.1.202 software with the curves tool. Images of colonies from different photos were then cut with the selection tool and moved to a new PNG file containing the cut colonies’ pictures on a transparent background. Using PALETTE WIZZARD 1.4, the color palette was extracted from the PNG file; then, the color palette was brought down to the 9 most characteristic color tones. The obtained palette was refined and color IDs were added with CYOTEK PALETTE EDITOR 1.7.0.411. Rayner’s color chart [46] was still used as a general guide for textual descriptions of colors, but we note that with the standard HEX color codes provided, the textual descriptions of the colors of colonies in the current study are not definitive and serve an indicative purpose.

*Pathogenicity tests*—Healthy hydroponically grown potato tubers, uniform in size, were washed and surface-sterilized in 0.5% sodium hypochlorite solution for 10 min, then rinsed in distilled water, air-dried, and peeled with a sterile blade (or left unpeeled in cases when a tuber was known to be clean coming from the sterile growing environment) and placed into sterile wet chambers. To determine the pathogenic status of the isolates, the tubers were wounded with a needle (sterile water for control or inoculated with a piece of mycelium taken from a margin of a 5-day-old colony grown on PDA). Three tubers were used for each strain for each of the two temperature conditions of 25 °C and 18 °C (to simulate storage conditions) for 25 days. For each isolate, a small infected-tissue sample was taken from the margin of the internal necrotic region with a sterile scalpel, surface-sterilized in 0.6% sodium hypochlorite for 15 s, rinsed twice in sterile distilled water, and blotted dry on sterile filter paper. The tissue pieces were then plated onto PDA Petri plates and incubated in the dark at 25 °C for 7 days. Following incubation, hyphal tips from the margins of actively growing isolates were removed with a sterile probe and plated onto one-fourth-strength PDA to generate pure cultures.

## 3. Results

### 3.1. Phylogeny

The concatenated alignment included 59 isolates representing 46 taxa. The alignment contained 1999 characters (including gaps) and was composed of three partitions: 852 characters for *rpb2*, 582 characters for *tub2*, and 559 characters for ITS. Of those, 1202 (60.13%) characters were constant, 701 were parsimony-informative, and 96 were parsimony-uninformative. For the Bayesian inference, the TIM + F+I + G4 model was selected as optimal for all partitions based on the results of the MrModeltest and IQ-tree tests. The results of ML and BI analyses of concatenated ITS*-tub2-rpb2* showed similar results that were used to build a phylogenetic tree (Figure 1). Phylogenetic analyses demonstrated that all studied isolates formed a monophyletic clade within the *Trichocladium* lineage. The clade is sister to *Trichocladium* sp. CBS 351.77, *T. crispatum*, and *T. atharcticum* but is separated with a high level of support (ML-BS 99%; MP-BS = 100%; PP = 1.00).

A significant level of similarity in all the studied isolates of *T. solani* was observed: 1825 (99.4%) characters were constant, with no parsimony-informative sites. No difference was found among ITS sequences whatsoever. Among *tub2* sequences, four transitions were determined at sites 31, 59, 62, and 122, and one transversion at site 129. Among *rpb2* sequences, all the isolates, apart from VKM F-4915, were identical, and VKM F-4915 had five transitions from G to A in sites 412, 429, 474, 484, and 490, and one deletion of A in site 454. Inside the clade, the best-fit model for Bayesian interference was determined to be JK. The best branching support for the mL tree built according to the model was 40% and 52% within the clade for overall alignment, with several internal branches showing near-zero (<0.0005) support. The obtained data suggest that the *T. solani* clade is monophyletic. 

Among the used genes, both *rpb2* and *tub2* demonstrated efficiency in the delimitation of *T. solani* from similar species having 320 and 294 parsimony-informative sites, respectively, and high resolution of the branches. The ITS region, on the contrary, had limited resolution and could not resolve *T. solani* clade from similar species with a satisfying level of support.

### 3.2. Pathogenicity towards Potato Tubers

All studied isolates showed pathogenicity to potato tubers in tests developing symptoms similar to those found in the tubers from which they were isolated, with darkened olivaceous areas of the infected tissue. There was no statistically significant difference in aggressiveness among isolates with damaged areas reaching 6–12 mm in diameter at 18 °C after 25 days and 15mm in diameter at 25–30 °C after 7 d (Figure 2). In all the experiments, the same cultures as inoculated, checked by morphology and sequencing, were isolated from infected potato tissues, confirming Koch’s postulates.

A supplementary experiment of inoculation of the whole tubers as described in [47] with both isolate VKM F-4915 and *Pseudomonas* sp. (culture provided “As Is” by the laboratory of molecular analyses of microorganisms, RUDN) caused a synergic effect resulting in the almost total destruction of a tuber’s tissues in the same time period of two weeks at 25 C.

### 3.3. Taxonomy

***Trichocladium solani*** sp. nov. Belosokhov & Elansky

MycoBank MB845967

*Typification*: Russia, Moscow region, commercial storage facility, from stored potato tubers, cultivar ‘Djura’ with symptoms of yellow rot on 3 April 2018, A.F. Belosokhov (VKM F-4902, **type**) GenBank: ITS = OL691125; *rpb2* = OM974683; *tub2* = OL314244. Russia, Moscow region, from the air from a potato storage facility on 28 July 2017, A.F. Belosokhov (VKM F-4903, paratype) GenBank: ITS = OL691126; *rpb2* = OL314249; *tub2* = OL314243. Russia, Moscow region, from potato tubers, cultivar ‘Gala’ with symptoms of yellow rot, 3 April 2018, A.F. Belosokhov (VKM F-4913, paratype) GenBank: ITS = OL691123; *rpb2* = OL314251; *tub2* = OL314246. Russia, Kaluga region, village Nizhnie Pryski in the Kozelsky district; from potato tubers, cultivar ‘Romano’ with symptoms of yellow rot, 14 May 2018, S.N. Elansky (VKM F-4904, paratype) GenBank: ITS = OL691122; *rpb2* = OM974684; *tub2* = OL314247. Russia, Moscow region, village Rogachevo of the Dmitrovsky district; from seed potato with no apparent damages, 14 September 2018, A.F. Belosokhov (VKM F-4905, paratype) GenBank: ITS = OL691124; *rpb2* = OL314250; *tub2* = OL314245. Russia, Moscow region, commercial storage facility, isolated from potato tubers, cultivar ‘La Strada’ with symptoms of yellow rot and physical blemishes, 6 December 2019, A.F. Belosokhov, (VKM F-4914, paratype) GenBank: ITS = OL691121; *rpb2* = OL314252; *tub2* = OL314248. Moscow region, potato tubers of cultivar ‘Gala’ with symptoms of yellow rot, March 2020, E.M. Chudinova, (VKM F-4915, paratype) GenBank: ITS = OL638311; *rpb2* = OL442086; *tub2* = OL674250.

*Etymology*: *solani* (Latin) refers to the first known host plant (*Solanum tuberosum* L.) from which isolates were obtained. 

*Diagnosis*: Colonies raised were cottony, citric, or luteous to pale; attaining 58 mm diam after 7 d at 25 °C on oatmeal agar; sometimes becoming olivaceous green with age. The soluble pigment was absent, reverse uncolored. Conidia absent, *Acremonium*-like conidiophores phialidic, formed laterally from hyphae, hyaline, unbranched, 16–18 μm long, 1.5–2.2 μm wide near the base. *Acremonium*-like conidia formed basipetally in chains, aseptate, obovoid, ellipsoidal, with a truncated base and a rounded apex, 1.8–2.1 × 3–3.2 μm; teleomorph unknown. Cellulolytic. Pathogenic to potato tubers, causing yellow rot.

*Description*: *Somatic hyphae* hyaline, 1.5–3 μm wide. *Asexual structures* appearing phialidic *Acremonium*-like conidiophores, formed laterally from the hyphae, hyaline, unbranched, (14)16–18(20) μm long, 1.5–2.5 μm wide near the base (Figure 3A). *Acremonium*-like conidia formed basipetally in chains that sometimes curl up (Figure 3B) so that at a later stage, spores might appear in heads. Spores aseptate, obovoid, ellipsoidal, with a truncated base and a rounded apex, (1.6)1.8–2.1(2.3) × (2.7)3–3.2(3.4) μm (n = 87, 7 specimens). Swollen chlamydospores, intercalary in hyphae, composed of hyaline, thick-walled subglobose to angular cells, (5)7–9(11) × (4)7–9(10.5) μm (n = 53, six specimens) which formed in branched or unbranched chains (Figure 3D,E), later appearing several subglobose cells in clusters (Figure 3E). *Teleomorph* not observed.

*Colony morphology*: (Figure 4) Colonies on OA were 48–67 mm in diam after 7 d at 25 °C; entire edge to slightly undulate, with a thick aerial mycelium; texture obverse cottony, pale to luteous, soluble pigment absent; reverse uncolored. Colonies on MEA were similar to those on OA with a slightly warmer shade of yellow, 46–63 mm diam after 7 d at 25 °C. Colonies on CMA 49–58 mm diam after 7 d at 25 °C; entire edge; aerial mycelium relatively thick, luteous to white; texture cottony to floccose; soluble pigment absent; reverse uncolored. Colonies on PCA 53–65 mm diam after 7 d at 25 °C entire; edge to undulate; aerial mycelium relatively thin, pale luteous to white, texture floccose with cottony center; soluble pigment absent; reverse uncolored. The full range of color variations on different media brought down to the nine most featured shades are demonstrated in Figure 5.

*Notes*: Descriptions of colonies are given for 7-day-old cultures grown on standard media. However, more saturated shades of yellow can be observed if grown on rich media with abundant nutrients or at a later stage of growth. Regardless of the medium, colonies can become thin floccose to adpressed with age and develop dark olivaceous-green soluble pigment.

*Ecology*: All studied isolates appear to be more or less thermophilic and able to utilize cellulose as a source of carbon. Observed isolates are pathogenic to potato tubers causing yellow rot during storage.

## 4. Discussion

### 4.1. Relations and Features of the Species

Phylogenetically, *Trichocladium solani* is shown to be close to *T. crispatum* (=*Chaetomium crispatum*) and *T. antarcticum* (=*Thielavia antarctica*) and falls into subclade 2 defined in [13] as containing only sexual species that produce ostiolate or non-ostiolate ascomata. The new species appear to have neither a teleomorph nor typical *Trichocladium-like* conidiation similar to *T. amorphum* X. Wei Wang & Houbraken, described in [13]. Numerous attempts were undertaken to induce sporulation among all of the studied isolates. All the methods described in the methodical part of the current study: UV-A light for 12 h described causing sporulation in fungi in [41,42]. Heat-shock treatment of 42 °C for 2h, mentioned by [43], or growth on Hutchinson medium [44] with different cellulose substrates as a source of carbon for 6.5 months have not triggered the development of a teleomorph or typical *Trichocladium*-like solitary, 1- to 2-celled, obovate, hyaline or dark monoblastic conidia. None have been observed in situ in the source potato tubers. To the best of our knowledge, nonspecific, *Acremonium*-like phialidic conidia are the only way of sporulation for the species and the absence of a teleomorph appears to be intrinsic. This feature, along with the phylogenetic placing, contradicts the connection suggested in [13] between the morphological features and phylogenetic subclades of the genus.

*Trichocladium solani* also appears to be closely related to *Trichocladium* sp. CBS 351.77, although the isolate was clearly placed apart from the species’ clade. CBS 351.77, collected by W. Gams in 1946 and identified by S. J. Hughes [48] as an unknown four-spored *Chaetomium,* has been reviewed in [13]. It was noted that the isolate was sterile, and the authors were unsure if it belonged to any of the species discussed in that study. *Trichocladium* sp. CBS 351.77 may be one of the close relatives or, perhaps, an ancestor form. However, we too are hesitant to place the isolate inside the *T. solani* until more data regarding the species’ geographical distribution and molecular diversity are collected in further studies.

### 4.2. Ecology

All known isolates of the species are related to potato tubers: the first isolate (VKM F-4903) for this study was obtained from an air sample from the sorting room of the potato storage facility. All other isolates are from infected potato tubers. In our experiments, all the studied isolates were pathogenic towards potato tubers damaging their tissue.

During the growth on the Hutchinson medium with different cellulose substrates, all the studied isolates of *T. solani* showed the ability to decompose cellulose. Among the sources of cellulose used in the study, the combination of oak and maple shavings appeared to be the most favorable for the isolates followed by ashless pulp, cellophane, and straw. Sulphated (kraft) cellulose and pine wood allowed the least abundant development of the mycelium. The ability to decompose cellulose has been listed as one of key pathogenicity factors for many plant-pathogenic fungi [49]. It is possible that the same enzymes used in the cellulose decomposition play a role in the lithic infection of the tuber cell wall. It is worth pointing out that the ability to utilize cellulose as a source of carbon is not uncommon among the Chaetomiaceae family [12,50]; however, the pathogenicity to living plants has not been usually reported for the group.

*Trichocladium solani* appears to be highly pathogenic towards potato tubers: the observed source tubers have usually suffered substantial damage for more than 50% of the mass from the infection that is normally accompanied by bacterial and nematode agents (Figure 6); although, in our experience, only 10–15% of the surface is where the damage is noticeable from the outside view. The infection tends to spread inwards causing extensive cavities inside a tuber. This feature can be attributed to the fact that the studied isolates tend to increase pathogenicity at higher temperatures in the experiments. Under the lab conditions, colder temperature caused a slow development of the infection whereas warmer temperatures noticeably improved the rate at which the tissue was colonized. We hypothesize that in the environment of potato storage, the rot spreads inwards along with bacteria, causing a local temperature rise which is better sustained in the microclimate inside the tubers’ cavities. Our current understanding of the infection suggests that it will likely manifest in storages with inefficient cooling conditions or ventilation or when stored tubers are not evenly distributed which leads to the development of clusters with increased local temperature and humidity [51]. A supplementary experiment with co-inoculation with a culture of *Pseudomonas* sp. hints that bacterial infections of tubers might significantly increase the rate of the rot on potato tubers in storages.

### 4.3. Monitoring of the Infection

The control of any infection starts with monitoring and identification. However, *Trichocladium solani* might prove challenging to reveal in routine storage surveys: infected potato tubers present symptoms that carry a strong resemblance to potato dry rot caused by fungi of the genus *Fusarium* [52,53]. The similarity of the two can most prominently be observed when comparing yellow rot symptoms caused by *T. solani* (Figure 6) with those of *Fusarium* dry rot presented by Wharton et al. [54]. In our experience, *T. solani* infection on tubers in their natural unwashed state in storage can be highly challenging to distinguish from the *Fusarium* dry rot, even by a trained specialist. Some thoroughly washed tubers in the late stages of the infection can have slight features that can point out the *T. solani* infection, such as the color of the overgrowing mycelium or shade of the necrosis. The difference is seen only in dissection where the mycelium beneath the necrosis and the infected tissues exhibit a yellowish, greenish, or olivaceous tint (Figure 6), in contrast to the mainly reddish or whitish tints associated with *Fusarium* infections [52], which, however, may also exhibit yellow tints thanks to the coloration of spores [55]. It is necessary to point out that differences in coloration of the lesions are mostly seen during the late stages of the infection. Therefore, symptomatic differentiation of the infections should not be carried out visually, as a final diagnosis and molecular identification or culture isolation are necessary to determine the etiologic agent of the infection. The new infection has been named “Yellow rot of the potato tubers”.

We would like to stress that the identification of the species is impossible by the ITS region, a popular barcode choice among various fungi. The internal transcribed spacer regions are insufficient and show little variability inside the Chaetomiaceae family [5,10,13], with *T. asperum* and *Humicola grisea* ITS sequences being identical [5]. Our obtained sequences of *T. solani* ITS regions also showed high similarity (99.65%, 99.82% and 99.47%, respectively) to *T. asperum*, *T. crispatum* and *H. grisea*. This fact adds to the challenge of identifying the yellow rot in tubers when use of highly specific genetic markers is unavailable.

Due to the absence of the more specific sequences of *tub2* and *rpb2* regions in databases before the work of Wang et al. [13] and the rare usage of molecular sequencing in potato monitoring surveys, it is difficult to determine the world distribution of *T. solani* and the spread of the yellow rot. However, to the best of our knowledge, no other *Trichocladium* species is pathogenic to, or closely associated with, potato-related substrates. Accepting the fact that the following observation is a purely hypothetical extrapolation, it can be noted that mentions of supposed *Trichocladium* can be encountered in some previous literature. Before the work of Wang at al [13], ITS sequences of *T. solani* were aligning with *T. asperum* MF782819.1 and *H. grisea* KU705826.1/*T. griseum* MN547382.1 via the BLAST algorithm, and after the mentioned paper, with *T. crispatum*/*Chaetomium crispatum* MH861360.1. Maintaining reasonable moderation in the conclusions that can be drawn out from this fact, we can observe that ‘*Trichocladium asperum*’ is mentioned from sclerotia of *Rhizoctonia solani* from a potato filed in Denmark in 1979 [56]; from eggplant (*Solanum melongena* L.) fruits in Saudi Arabia in 1985 [57]; from the soil after three continuous years of potato cultivation in Poland in 2002 [58]; then, again, in Poland in 2004 from the potato plant’s rhizosphere [59]; from potato tubers in France in 2010 [60]; and then from potato tubers in storage in Latvia in 2015 [61]. ‘*Trichocladium* sp.’ is mentioned on potato tubers in Poland in 1987 [62], and from soil after a long-term period of the cultivation of *Solanum tuberosum* in Gansu province in China in 2017 [63]. This observation is in no way proving anything about the true geographic range of *T. solani* on potato tubers in the world. However, it hints that the real distribution of the species is likely not limited to Russia where isolates were initially found. The development of a reliable and specific test system for *T. solani* and extensive research is required to monitor and control the spread of yellow rot in storages around the world.

## 5. Conclusions

*Trichocladium solani*, the newly described species, is a pathogen of potato tubers. It was described in those from Russia; however, circumstantial evidence suggests a wider distribution of the species is likely to be confirmed in the following studies. The new species is highly pathogenic to potato tubers and causes yellow rot of potato tubers. As a result of this cellulolytic activity, the fungus appears to cause significant damage to the tuber during storage. The pathogen can be tricky to distinguish from a dry rot caused by *Fusarium* spp. and produces few distinct morphological features: it has no observed teleomorph and, morphologically, is characterized by nonspecific *Acremonium*-like conidia on single phialides and chains of swollen chlamydospores. Internal transcribed spacer regions (ITS) were found to be insufficient for species delimitation and cannot be used to distinguish the species from other *Chaetomiaceae* in screening studies. The development of a species-specific test system and extensive potato survey is the next necessary step in the assessment of the world distribution of the pathogen and the development of control measures towards the infection.

## Figures and Tables

**Figure 1 jof-08-01160-f001:**
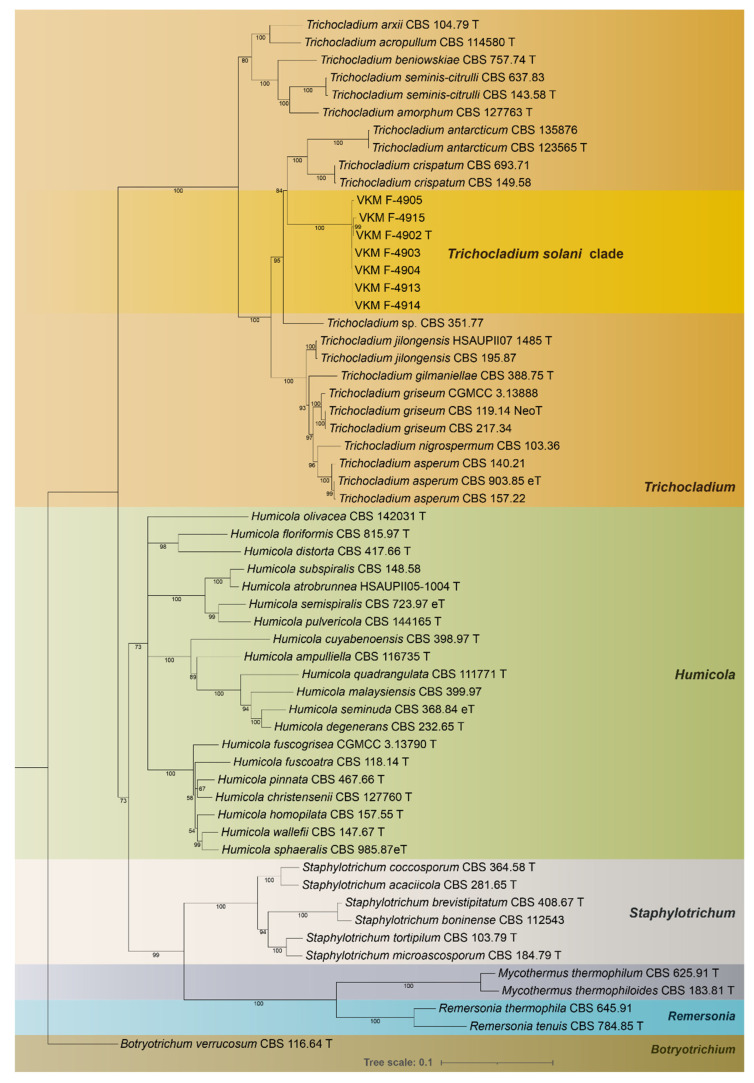
Phylogenetic tree inferred from a maximum-likelihood analysis of the concatenated alignment including the partial rpb2 gene region, partial tub2 gene region, and ITS region. The confidence values are indicated at the branches. The scale bar shows the expected number of changes per site.

**Figure 2 jof-08-01160-f002:**
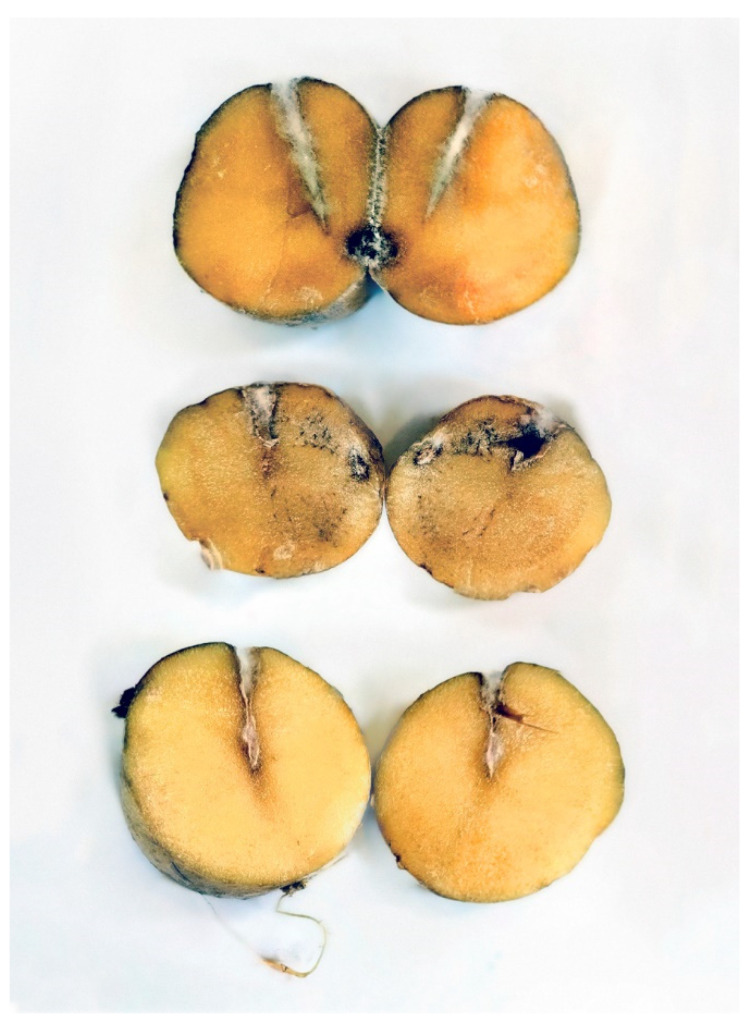
Cross-section of potato tubers infected with *Thichocladium solani* strains from top to bottom: VKM F-4904, VKM F-4902, and VKM F-4913 after 7 days at 25 °C.

**Figure 3 jof-08-01160-f003:**
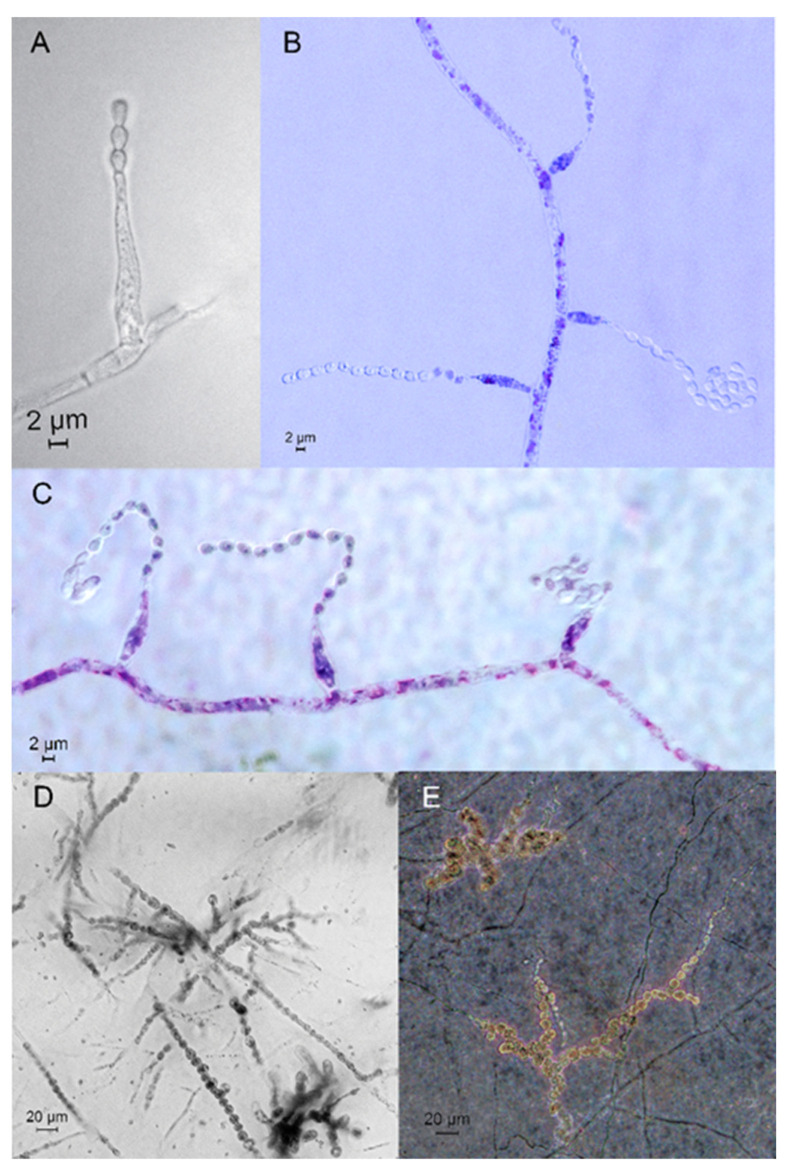
Microscopic features of *Trichocladium solani* sp. nov. (**A**)—Acremonium-like phialide. (**B**,**C**)—*Acremonium*-like sporulation (stained with toluidine blue). (**D**,**E**)—branched chains of chlamydospores.

**Figure 4 jof-08-01160-f004:**
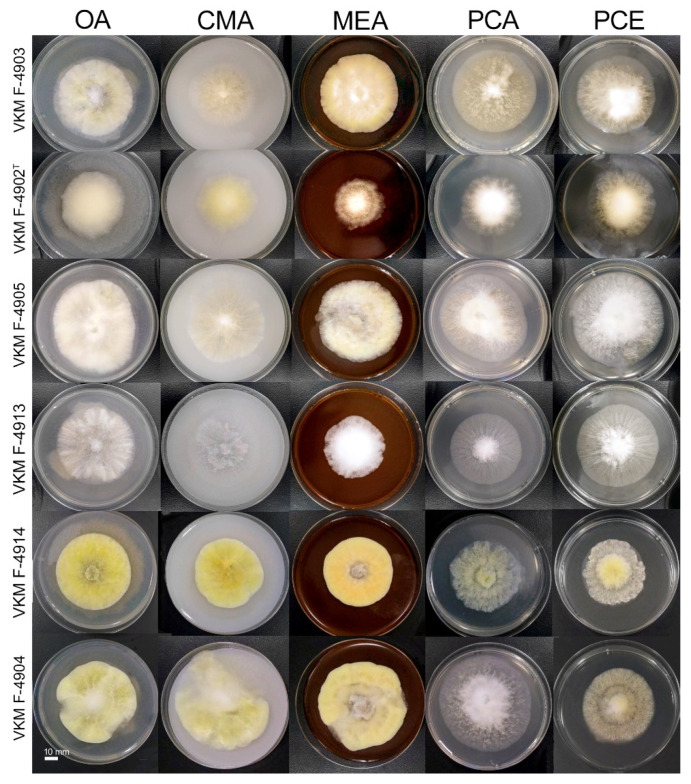
*Trichocladium solani* sp. nov. on different media. OA—oatmeal agar; CMA—cornmeal agar; MEA—malt extract agar; PCA—potato carrot agar; PCE—potato carrot extract. Cultures were incubated for 7 days at 25 °C.

**Figure 5 jof-08-01160-f005:**
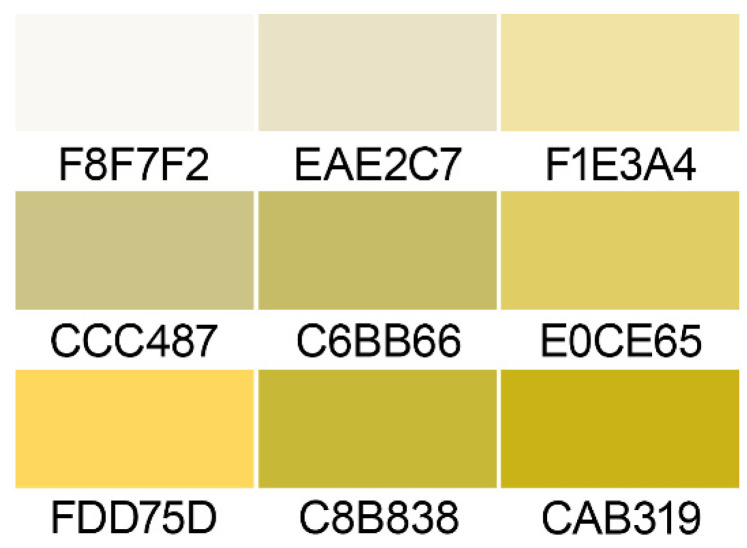
The spectrum of colors of colonies brought down to the 9 most prevalent shades.

**Figure 6 jof-08-01160-f006:**
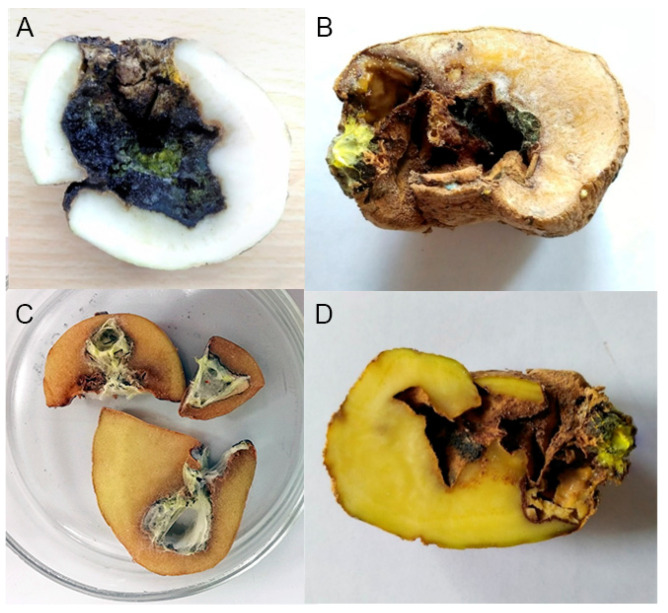
Source potato tubers, infected by yellow rot. (**A**) VKM F-4914, (**B**) VKM F-4913, (**C**) VKM F-4915, (**D**) VKM F-4902.

**Table 1 jof-08-01160-t001:** Details of strains included in the study. Strains isolated in the study are formatted with bold font. ^T^—type material; ^eT^—ex-type.

Species	Culture Accession Number	Origin	GeneBank Accession Numbers		
			ITS	*rpb2*	*tub2*
*Trichocladium solani*	**VKM F-4903**	Air from potato storage facility, Moscow region, Russia. July 2017	OL691126	OL314249	OL314243
	**VKM F-4902** ^T^	Potato tubers, cultivar ‘Djura’ with symptoms of yellow rot. Moscow region, Russia. March 2018	OL691125	OM974683	OL314244
	**VKM F-4913**	Potato tubers, cultivar ‘Gala’ with symptoms of yellow rot. Moscow region, Russia. March 2018	OL691123	OL314251	OL314246
	**VKM F-4904**	Potato tubers, cultivar ‘Romano’ with symptoms of yellow rot. Kaluga region, Russia. May 2018	OL691122	OM974684	OL314247
	**VKM F-4905**	Potato tubers, cultivar ‘Gala’ with symptoms of yellow rot. Moscow region, Russia. September 2018	OL691124	OL314250	OL314245
	**VKM F-4914**	Potato tubers, cultivar ‘La Strada’ with symptoms of yellow rot. Moscow region, Russia. November 2019	OL691121	OL314252	OL314248
	**VKM F-4915**	Potato tubers of cultivar ‘Gala’ with symptoms of yellow rot. Moscow Region. March 2020	OL638311	OL442086	OL674250
*Botryotrichum verrucosum*	CBS 116.64 ^T^	Salt marsh soil, mature dunes, UK. January 1962	LT993567	LT993486	LT993648
*Humicola ampulliella*	CBS 116735 ^T^	Discarded sock, China. August 2013	LT993568	LT993487	LT993649
*Humicola atrobrunnea*	HSAUPII05-1004 ^T^	Soil, China, Guizhou	LT993570	LT993489	LT993651
*Humicola christensenii*	CBS 127760 ^T^	Soil, USA, Minnesota. 1988	LT993571	LT993490	LT993652
*Humicola* *cuyabenoensis*	CBS 398.97 ^T^	Rain forest, Ecuador. March 1993	LT993573	LT993492	LT993654
*Humicola degenerans*	CBS 232.65 ^T^	Soil under mixed forest, Canada, Ontario. August 1964	LT993574	LT993493	LT993655
*Humicola distorta*	CBS 417.66 ^T^	*Populus tremuloides* dead leaf, USA, Iowa	LT993577	LT993496	LT993658
*Humicola floriformis*	CBS 815.97 ^T^	Fallen leaves, Thailand, Sukhothai	LT993578	LT993497	LT993659
*Humicola fuscoatra*	CBS 118.14 ^T^	Soil, Norway. 1914	LT993579	LT993498	LT993660
*Humicola fuscogris* *ea*	CGMCC 3.13790 ^T^	Soil, China, Shennongjia	LT993581	LT993500	LT993662
*Humicola homopilata*	CBS 157.55 ^T^	Filter paper in soil, Norway	LT993582	LT993501	LT993663
*Humicola leptodermospora*	CBS 120095 ^T^	Forestal soil, Brazil	LT993584	LT993503	LT993665
*Humicola m* *alaysiensis*	CBS 399.97	*Elaeis guineensi*, Mylaysia, Selangor. March 1974	LT993586	LT993505	LT993667
*Humicola mutabilis*	CBS 779.71 ^T^	Soil, Israel	LT993588	LT993507	LT993669
*Humicola olivacea*	CBS 142031 ^T^	Dust, USA	LT993589	LT993508	LT993670
*Humicola pinnata*	CBS 467.66 ^T^	Dead wood, USA, Coronado National Forest	LT993590	LT993509	LT993671
*Humicola pulvericola*	CBS 144165 ^T^	Dust, Mexico	LT993591	LT993510	LT993672
*Humicola quadrangulata*	CBS 111771 ^T^	Soil, Brazil	LT993593	LT993512	LT993674
*Humicola seminuda*	CBS 368.84 ^eT^	Soil, Canada, Ontario. April 1981	LT993594	LT993513	LT993675
*Humicola semispiralis*	CBS 723.97 ^eT^	Paper, Canada, Toronto	LT993597	LT993516	LT993678
*Humicola sp* *haeralis*	CBS 985.87 ^eT^	Soil, France	LT993598	LT993517	LT993679
*Humicola subspiralis*	CBS 148.58	Leaf fragments in soil, China	LT993599	LT993518	LT993680
*Humicola wallefii*	CBS 147.67 ^T^	Soil, Zaire	LT993602	LT993521	LT993683
*Mycothermus thermophiloides*	CBS 183.81 ^T^	Soil, USA, Indiana	LT993603	LT993522	LT993684
*Mycothermus thermophilum*	CBS 625.91 ^T^	Chicken nest straw, USA, Nevada	LT993604	LT993523	LT993685
*Remersonia tenuis*	CBS 784.85 ^T^	Dung of horse, India	LT993609	LT993528	LT993690
*Remersonia thermophila*	CBS 645.91	Compost, Netherlands	LT993611	LT993530	LT993692
*Staphylotrichum acaciicola*	CBS 281.65 ^T^	Acacia karroo leaf litter, South Africa	LT993613	LT993532	LT993694
*Staphylotrichum boninense*	CBS 112543	Leaf litter, Brazil	LT993617	LT993536	LT993698
*Staphylotrichum brevistipitatum*	CBS 408.67 ^T^	*Eucalyptus* leaf litter, South Africa	LT993619	LT993538	LT993700
*Staphylotrichum coccosporum*	CBS 364.58 ^T^	Soil, Zaire	LT993620	LT993539	LT993701
*Staphylotrichum microascosporum*	CBS 184.79 ^T^	Soil from Mangifera orchard, Sudan	LT993624	LT993543	LT993705
*Staphylotrichum tortipilum*	CBS 103.79 ^T^	Dung of pine vole, USA, North Carolina	LT993625	LT993544	LT993706
*Trichocladium acropullum*	CBS 114580 ^T^	Soil, China	LT993626	LT993545	LT993707
	CBS 127763 ^T^	Greenhouse Soil, USA	LT993628	LT993547	LT993709
*Trichocladium antarcticum*	CBS 135876	*Usnea* cf. *aurantio-atra*, Antarctica	LT993630	LT993549	LT993711
	CBS 123565 ^T^	*Usnea* cf. *aurantio-atra*, Antarctica	LT993629	LT993548	LT993710
*Trichocladium arxii*	CBS 104.79 ^T^	Dung of kangaroo rat, USA, California	LT993631	LT993550	LT993712
*Trichocladium asperum*	CBS 903.85 ^eT^	Acidic soil, Germany	LT993632	LT993551	LT993713
	CBS 140.21	Unknown substrate, Netherlands	LT993633	LT993552	LT993714
	CBS 157.22	Unknown substrate, unknown country	LT993634	LT993553	LT993715
*Trichocladium beniowskiae*	CBS 757.74 ^T^	Grass, India	LT993635	LT993554	LT993716
*Trichocladium crispatum*	CBS 149.58	Estuarine sediment, Germany	LT993636	LT993555	LT993717
	CBS 693.71	Agricultural soil, Netherlands	LT993637	LT993556	LT993718
*Trichocladium gilmaniellae*	CBS 388.75 ^T^	Salt marsh soil, Kuwait	LT993638	LT993557	LT993719
*Trichocladium griseum*	CBS 119.14 NeoT	Soil, Norway	LT993639	LT993558	LT993720
	CBS 217.34	Unknown substrate, Germany	LT993640	LT993559	LT993721
	CGMCC 3.13888	Soil, China, Jilin	LT993641	LT993560	LT993722
*Trichocladium jilongensis*	HSAUPII07 1485 ^T^	Mountain soil, China, Tibet	LT993642	LT993561	LT993723
	CBS 195.87	Field soil, Germany	LT993643	LT993562	LT993724
*Trichocladium nigrospermum*	CBS 103.36	Meal, Netherlands	LT993644	LT993563	LT993725
*Trichocladium seminis-citrulli*	CBS 143.58 ^T^	Dung of fox, Turkmenistan	LT993645	LT993564	LT993726
	CBS 637.83	Dung of goat, Israel	LT993646	LT993565	LT993727
*Trichocladium sp.*	CBS 351.77	Wheat field soil, Germany	LT993647	LT993566	LT993728

## Data Availability

Newly generated ITS, TUB, and RPB1 sequences are deposited in GenBank under accession numbers specified in Table 1. All alignments are available provided as Appendix A.

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
