# Peer review of "Trichocladium solani sp. nov.—A New Pathogen on Potato Tubers Causing Yellow Rot"

_jof, 2022, doi:10.3390/jof8111160_

Round 1

Reviewer 1 Report

The work aims to present the identification of a new potato pathogen species belonging to the genus Trichocladium capable of infecting tubers during storage.

Both molecular and morphological methods were followed for identification, to which a slight physiological and pathogenetic characterisation was added.

The manuscript does not state the botanical species of the plant from which the pathogen was isolated. is it Solanum tuberosum?

I have my doubts about yellow rot. Could it be due to the pigmentation of the tuber flesh?

Therefore, I suggest changing the title to: Trichocladium solani sp. nov. - a new pathogen of Solanum tuberosum causing soft rot of tubers

Rewrite the abstract being more consistent and removing the phrases L12-13 and L15-17

The writing is not scientifically rigorous and there are repeated concepts. Be consistent.

In the Introduction, before subsection L39 it is necessary to talk about the plant species , importance, problems, susceptibility, postharvest ... etc.

The information about Chaetomiaceae L39-56 and genus Trichocladium L57-72 only needs to be mentioned in the Introduction, whereas it should be included more concisely in the discussion.

in M&M indicate the name of the botanical species of the isolation. L82 indicate how many tubers were collected and the incidence of symptoms for each locality

Tab. 1 and Text, why are some culture accession numbers highlighted?

the discussion is too long and should be schematised according to sub-paragraphs/bullet points: and indicate the importance of the genus Trichocladium among agricultural crops as a pathogen; the symptoms caused; whether there are other cases of Trichocladium implicated in post-harvest diseases

if parallels are drawn with fusarium dry rot, it would be good to show the two photos in comparison

Author Response

Thank you kindly for your review of the manuscript. Here is the point-by-point reply to your comments: 

The manuscript does not state the botanical species of the plant from which the pathogen was isolated. is it Solanum tuberosum?

Fixed. Now botanical species of the host plant is stated more explicitly in the text.

I have my doubts about yellow rot. Could it be due to the pigmentation of the tuber flesh? Therefore, I suggest changing the title to: Trichocladium solani sp. nov. - a new pathogen of Solanum tuberosum causing soft rot of tubers

The name for the infection is given strictly for the tint of the necrotized area that is mostly dark-olivaceous with areas of yellow mycelium. The name does not imply that the living flesh of a tuber is affected by color change. The yellow flesh of the tuber in Figure 6.D. is due to the features of the variety 'Djura'. We changed L19-20 so it is now more precise to avoid confusion.

Rewrite the abstract being more consistent and removing the phrases L12-13 and L15-17

The abstract has undergone revision

The writing is not scientifically rigorous and there are repeated concepts. Be consistent.

Please, see the revised version of the manuscript. If the issue remains, could you please, specify inconsistencies?

In the Introduction, before subsection L39 it is necessary to talk about the plant species , importance, problems, susceptibility, postharvest ... etc.

We added the parapgaph on that after the introduction to the current state of the taxon. See L74-87

The information about Chaetomiaceae L39-56 and genus Trichocladium L57-72 only needs to be mentioned in the Introduction, whereas it should be included more concisely in the discussion.

We appreciate the suggestion, however, our stand is that the manuscript is primarly about the new fungal species and the study belongs into the taxonomy first and the plant pathology second. Therefore, we believe that it is essential to focus on the species' relationships and position within the group as well as the group's definitive features so the introduction have not been revised in that regard.

in M&M indicate the name of the botanical species of the isolation. L82 indicate how many tubers were collected and the incidence of symptoms for each locality

Done

Tab. 1 and Text, why are some culture accession numbers highlighted?

As it is specified in L109, highlighted in bold are the strains, isolated in the current study. Highlighted with cyan are strains that haven't been deposited in the public collection yet. We are currently in correspondence with the CBS collection to arrange the deposition of the remained strains.

the discussion is too long and should be schematised according to sub-paragraphs/bullet points

Done. However, we are not sure if this change was felicitous

indicate the importance of the genus Trichocladium among agricultural crops as a pathogen; the symptoms caused; whether there are other cases of Trichocladium implicated in post-harvest diseases

Please, see L377 and L430-432

if parallels are drawn with fusarium dry rot, it would be good to show the two photos in comparison

We agree, the comparison would be an effective mean of delivering the idea. However, we fear that directly presenting a photo from a third-party source may cause copyright issues. Therefore we do not include any additional figures, but we revised L400-402 so it is more now more clearly implies the invitation to see the referenced source.

Reviewer 2 Report

Dear authors and editors,

Here is the review of the manuscript titled "Trichocladium solani sp. nov. – a new pathogen on potato tubers causing yellow rot" written by Arseniy  Belosokhov & co-authors.

The aim was to describe Trichocladium solani, as a new species to science from Russia. The species has no observed teleomorph. Morphologically, it is characterized by Acremonium-like conidia on single phialides and chains of swollen chlamydospores. Phylogenetic analysis, based on three gene regions (ITS, tub2, and rpb2) placed the new species in a monophyletic clade within the genus Trichocladium with high support. The new species is a pathogen of potato tubers causing lesions similar to Fusarium.

The methods used and analyses performed are appropriate for this kind of study. Morphological description, phylogenetic study, and discussion are exhaustive and cover all needed parts. The English language is mostly well. The authors followed the newest version of International code of nomenclature for algae, fungi, and plants. Mycobank number should be added and references should be numbered throughout the text.

All my remarks/suggestions on the manuscript are included in the pdf document attached. Other than that, the paper could be accepted for publication in JoF.

Best, Reviewer

Author Response

Thank you kindly for your most helpful review and comments! We sincerely appreciated your work with the manuscript.
Please, see the changes in the manuscript and replies to your comments below:

  • All suggestions regarding the improvement of the text/phrasing have been accepted and applied.
  • Missing information has been added
  • Updated the numeration of figures
  • A few changes to the text were made in accordance with the recommendations of the first reviewer. Mostly L77-90 and L408-409

Replies to comments:
Are all sequences used in this study other than yours generated by Wang & al.[13]?. Or Wang & al. used sequences generated from other sources?
And the related comment: Very important issue: Please insert the additional column in the table and cite all the original references that produced sequences used in this study. (L90)

Yes, all sequences that were not generated in this study were produced by Wang et al, 2019 [13], so we felt that an additional column would be excessive in this case. L111 should now more clearly indicate the case to avoid future confusion.

Add "Figures 3 & 4" after the names of the species authors.

We agree with your recommendation, however, we did not change the placing of figures 3&4 in the file due to the extreme chaos it causes. Instead, we asked the editorial office to make the change for us before the print (should the manuscript be accepted for publication), so it is done by a professional layout manager.